# Robotic versus laparoscopic total mesorectal excision with lateral lymph node dissection for advanced rectal cancer: A systematic review and meta-analysis

**Mohamed Ali Chaouch**[1‡]*, **Mohammad Iqbal Hussain**[2‡], **Adriano Carneiro da Costa**[3], **Alessandro Mazzotta**[4], **Bassem Krimi**[5], **Amine Gouader**[5], **Eddy Cotte**[6], **Jim Khan**[2], **Hani Oweira**[7]

1 Department of visceral and digestive surgery, Monastir University Hospital, Monastir, Tunisia, 2 Department of Robotic Colorectal Surgery, Portsmouth Hospitals University NHS Trust, Portsmouth, United Kingdom, 3 Department of Surgery, Federal University of Pernambuco, Recife, Pernambuco, Brazil, 4 Department of Surgery, M. G. Vannini Hospital, Istituto Figlie Di San Camillo, Rome, Italy, 5 Department of Surgery, Perpignan Hospital Center, Perpignan, France, 6 Department of Visceral Surgery, University Hospital of Lyon, University of Lyon, Lyon, France, 7 Department of Surgery, Universitäts Medizin Mannheim, Heidelberg University, Mannheim, Germany

‡ MAC and MIH are contributed equally as joint first authors
* docmedalichaouch@gmail.com

**Data Availability Statement:** All relevant data are within the article.

## Abstract

### Introduction

Lateral pelvic node dissection (LPND) poses significant technical challenges. Despite the advent of robotic surgery, determining the optimal minimally invasive approach remains a topic of debate. This study aimed to compare postoperative outcomes between robotic total mesorectal excision with LPND (R-LPND) and laparoscopic total mesorectal excision with LPND (L-LPND).

### Methods

This meta-analysis was conducted according to the Preferred Reporting Items for Systematic Review and Meta-analysis (PRISMA) 2020 and AMSTAR 2 (Assessing the Methodological Quality of Systematic Reviews) guidelines. Utilizing the RevMan 5.3.5 statistical package from the Cochrane Collaboration, a random-effects model was employed.

### Results

Six eligible studies involving 652 patients (316 and 336 in the R-LPND and L-LPND groups, respectively) were retrieved. The robotic approach demonstrated favourable outcomes compared with the laparoscopic approach, manifesting in lower morbidity rates, reduced urinary complications, shorter hospital stays, and a higher number of harvested lateral pelvic lymph nodes. However, longer operative time was associated with the robotic approach. No significant differences were observed between the two groups regarding major

**Funding:** The author(s) received no specific funding for this work.

**Competing interests:** No conflict of interest to disclose

complications, anastomotic leak, intra-abdominal infection, neurological complications, LPND time, overall recurrence, and local recurrence.

## Conclusions

In summary, the robotic approach is a safe and feasible alternative for Total Mesorectal Excision (TME) with LPND in advanced rectal cancer. Notably, it is associated with lower morbidity, particularly a reduction in urinary complications, a shorter hospital stay and increased number of harvested lateral pelvic nodes. The trade-off for these benefits is a longer operative time.

## Introduction

Controversies exist regarding the management of rectal cancer with lateral pelvic node (LPN) involvement. In the West, it is considered a systemic disease, whereas in the eastern part of the world, it is considered a regional disease [1]. The West is managed with neoadjuvant chemoradiotherapy followed by total mesorectal excision (TME) [1]. In the East, promising results have been demonstrated when LPN metastasis (LPNM) is treated with TME and LPN dissection (LPND) [1,2]. Furthermore, a recent collaborative multicenter LPN study concluded that TME with LPND decreased local recurrence in patients with suspected LPN metastases, even after preoperative chemoradiotherapy [3]. LPND is a challenging procedure due to its technical difficulties and the high risk of surgical morbidity. Despite this, LPND is widely performed using open and laparoscopic approaches [4]. A recent meta-analysis [4] concluded that laparoscopic LPND may be a better alternative to conventional open LPND for advanced rectal cancer with lower postoperative morbidity and shorter postoperative hospital stay. The laparoscopic approach poses challenges due to limitations such as a restricted field of vision, a flat 2-dimensional view of the narrow pelvis, and constrained dexterity. Robotic technologies have been employed to address these deficiencies. This technological advancement also presents opportunities to extend LPND. Despite the potential advantages, there is a scarcity of studies directly comparing the outcomes between Robotic Total Mesorectal Excision (RTME) with LPND and Laparoscopic Total Mesorectal Excision (LTME) with LPND, and the existing findings are inconsistent. By synthesizing the available evidence, this study endeavors to contribute to the highest level of evidence in the literature on this subject. Critical evaluation of postoperative outcomes will shed light on the comparative effectiveness and potential advantages of these two surgical approaches, offering valuable insights for clinical decision-making in the context of rectal cancer surgery.

This systematic review and meta-analysis aim to provide a comprehensive comparison of postoperative outcomes between RTME with LPND and LTME with LPND.

## Methods

This study involved only human participants. This was a retrospective analysis of published cases that did not require informed consent. Ethical approval and consent to participate were not included in this review. The meta-analysis was conducted in accordance with the PRISMA 2020 (Preferred Reporting Items for Systematic Review and Meta-analysis) guidelines [5]. To evaluate its quality, we employed the AMSTAR 2 (Assessing the Methodological Quality of

Systematic Reviews) 2 tool [6]. The study protocol was registered in PROSPERO under the identification number CRD42023403909.

## Electronic database searches

We conducted a comprehensive electronic literature search through April 15, 2023, without language restrictions. The search encompassed multiple databases, including the Cochrane Library's Controlled Trials Registry and Database of Systematic Reviews, PubMed/MEDLINE from the United States National Library of Medicine, Google Scholar, Excerpta Medica Database (Embase), and Scopus. The keywords employed in the search strategy were "Randomized Controlled Trials," "Clinical Controlled Trials," "lateral lymph nodes dissection," "total mesorectal excision," "rectal cancer," "robotic," "laparoscopy," "postoperative morbidity," "mortality," "outcomes," "overall survival," "disease-free survival," and "neoplasm recurrence." In addition, we manually reviewed the reference lists of the retrieved articles to identify relevant clinical trials.

## Eligibility criteria

**Studies:** We included all randomized controlled trials (RCTs) and controlled clinical trials (CCTs) that compared robotic Total Mesorectal Excision (RTME) with Laparoscopic LPND, and laparoscopic Total Mesorectal Excision (LTME) with LPND. Non-comparative studies, editorials, letters to editors, review articles, and case series were excluded from analysis.

**Population:** This study focused on adults of any sex who underwent either RTME or LTME with LPND for advanced rectal cancer.

**Intervention Group:** Participants who underwent RTME with LPND for advanced rectal cancer were categorized into the R-LPND group.

**Control Group:** Patients who underwent LTME with LPND for advanced rectal cancer comprised the L-LPND group.

**Outcomes:** The primary outcome assessed was the morbidity rate, with secondary outcomes including anastomotic leak, intra-abdominal infection, urinary complications, operative time, hospital stay, number of Laparoscopic Pelvic Nodes (LPN) harvested, LPND time, overall recurrence, and local recurrence. Mortality and morbidity were considered when they occurred within 90 and 30 days after rectal resection, respectively.

**Study Selection:** Following independent literature searches conducted by two authors, all abstracts were independently reviewed. The inclusion criteria considered RCTs and CCTs were included in this study. The full texts of studies meeting these criteria were retrieved, and any disagreements were resolved through discussion with a third review team member.

**Assessment of Study Quality and Risk of Bias:** Two authors independently evaluated the selected studies based on predetermined criteria. Quality assessment for CCTs and RCTs used the Methodological Index of Non-Randomized Studies (MINORS) [7] and the Consolidated Standards of Reporting Trials (CONSORT) statement [8], respectively. Studies scoring below 13 on the MINORS or CONSORT scale were excluded because of fair quality. The risk of bias in RCTs was assessed using the Cochrane tool for bias assessment (RoB2) [9], whereas the risk of bias in CCTs was evaluated using the Newcastle Ottawa Scale (NOS) [10].

**Missing Data:** In instances of unclear bias domains or missing primary outcome information, the authors were contacted via email. If data were not numerically reported, information was extracted from the figures.

**Handling Continuous Data:** Continuous data were analyzed using the Review Manager 5.3.5 statistical package from the Cochrane collaboration for meta-analysis [11]. When the mean and standard deviation (SD) were not provided, they were estimated from the interquartile range (IQR) and median following the formula described by Hozo et al. [12].

**Assessment of Heterogeneity:** Heterogeneity was assessed using the Cochrane $Chi^2$ test (Q-test), $I^2$ statistic, and variance $Tau^2$ to estimate the degree of heterogeneity [13]. Funnel plots were used to identify studies responsible for heterogeneity, and subgroup analysis was conducted when all the included studies reported outcomes.

**Summary of findings:** Two authors independently evaluated the certainty of evidence using The Grading of Recommendations Assessment, Development, and Evaluation (GRADE) [14]. The factors considered included study limitations, consistency of effect, imprecision, indirectness, and publication bias. The certainty of evidence was classified as high, moderate, low, or very low. Criteria for upgrading certainty included a large effect, dose-response gradient, and a plausible confounding effect. The Cochrane Handbook for Systematic Reviews of Interventions (sections 8.5 and 8.7, and chapters 11 and 12) and GRADEpro GDT software were utilized to prepare 'Summary of Findings' tables, providing explanations for downgrading or upgrading certainty using footnotes with comments.

**Evaluation of effect size:** Meta-analysis was performed using the RevMan 5.3.5 statistical package from the Cochrane Collaboration [11]. The mean difference (MD) was selected as the effective measure for continuous data, whereas odds ratios (OR) with 95% confidence intervals (95% CI) were calculated for dichotomous variables. A random-effects model was applied, with a significance set at 0.05.

## Results

**Literature search results**: Nine potentially relevant articles (**Fig 1**). We retained six eligible studies [2,15–19] and three studies were excluded for the following reasons: a systematic review and meta-analysis comparing open and laparoscopic LPND [4], one review article comparing laparoscopic and robotic LPND [20], and one non-comparative study [21]. The articles were published between 2018 and 2023. All of these studies were Asian: three studies from South Korea, two from Japan, and one from China. There have been no RCTs on this subject. These studies included 652 patients (316 patients with R-LPND and 336 patients with L-LPND) (**Table 1**). The demographic data of the included studies are summarized in **Table 2.** The mean ages ranged between 57 and 63 years in the R-LPND group and 58.3 and 63 years in the L-LPND group. The sex ratio was 1.68 with a male predominance. The BMI ranged between 21.1 $kg/m^2$ and 23.4 $kg/m^2$ in the R-LPND group and between 22.8 $kg/m^2$ and 23.8 $kg/m^2$ in the L-LPND group. Tumor height ranged from 4 to 5 cm in both groups. The tumor size ranged from 2.9 cm to 4.5 cm in the R-LPND group and 3.1 cm to 4.5 cm in the L-LPND group. The follow-up duration among the different studies ranged from one to 44.6 months.

### Morbidity

The morbidity rate has been reported in six studies [2,15–19]. post-operative complications were reported in 77 of 316 patients in the R-LPND group and 130 of 336 patients in the L-LPND group. There was a lower morbidity rate in the R-LPND group than in the L-LPND group (OR = 0.52, 95%CI [0.34 to 0.79], p = 0.003] (**Fig 2A**) There was low heterogeneity among studies ($Tau^2$ = 0.08).

### Anastomotic leak

The anastomotic leak rate has been reported in six studies [2,15–19]. It was reported in 16 of 316 patients in the R-LPND group and in 27 of 336 patients in the L-LPND group. There was no significant difference between the two groups (OR = 0.74, 95%CI [0.36, 1.54], p = 0.42) (**Fig 2B**). There was low heterogeneity among studies ($Tau^2$ = 0.02).

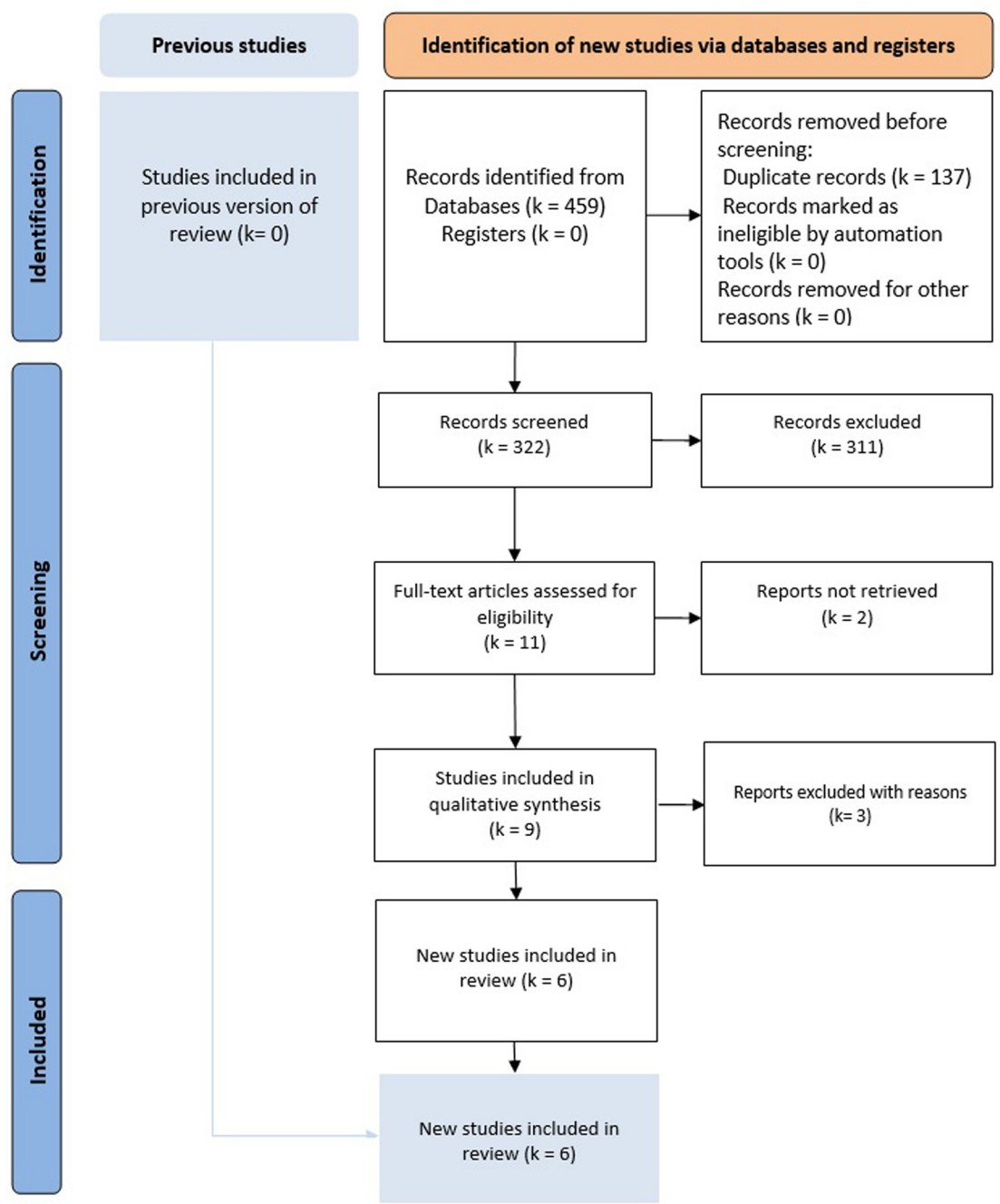

**Fig 1. Flow-diagram of the bibliographic research.**

### Intra-abdominal infection

The intra-abdominal infection rate was reported in four studies [16–19]. It was reported in 11 of 255 patients in the R-LPND group and eight in of 229 patients in the L-LPND group. There was no significant difference between the two groups (OR = 0.132, 95%CI [0.5 to 3.47], p = 0.55) (**Fig 2C**).

**Table 1. List of the retained studies.**

| First author | Country of origin | Journal | Year of publication | Study period | Study type | Study design | Number of patients* (Robo/Lap) | Quality assessment (MINORS) | Risk of bias assessment (NOS) |
|---|---|---|---|---|---|---|---|---|---|
| Bae | South Korea | Biomedicines | 2023 | 2015.2021 | Retrospective comparative | multicenter | 34/74 | 20 | 6 |
| Ishizaki | Japan | Techniques in Coloproctology | 2023 | 2013–2022 | Retrospective comparative | monocentric | 27/33 | 20 | 6 |
| Kim | South Korea | Surgical Endoscopy | 2018 | 2006–2014 | Retrospective comparative | monocentric | 50/35 | 16 | 5 |
| Morohashi | Japan | Surgical Endoscopy | 2020 | 2014–2020 | Retrospective comparative | monocentric | 40/55 | 18 | 6 |
| Song | South Korea | Tech Coloproctol | 2021 | 2006–2016 | Retrospective comparative | monocentric | 70/29 | 18 | 6 |
| Zhang | China | Surgical Endoscopy | 2023 | 2015–2021 | Retrospective comparative | multicentric | 95/110 | 20 | 6 |

## Urinary complications

The urinary complication rate was reported in six studies [2,15–19]. It was reported in 26 of 316 patients in the R-LPND group and in 62 of 336 patients in the L-LPND group. There was a lower rate of urinary complications in the two R-LPND groups (OR = 0.39, 95%CI [0.24 to 0.64]; p = 0.0002) (**Fig 2D**).

## Neurological complications

The neurological complication rate was reported in six studies [2,15–19]. It was reported in five out of 316 patients in the R-LPND group and 13 out of 336 patients in the L-LPND group. There was no significant difference between the two groups (OR = 0.47, 95%CI [0.18 to 1.22], p = 0.12) (**Fig 2E**).

**Table 2. Demographic data of the different retained studies.**

| Studies | Age (years) Robo/Lap | Gender (Male:female) Robo/Lap | BMI (mean, kg/m2) Robo/Lap | Rectal tumor location | | Tumor height (mean or median, cm) Robo/Lap | ASA score Robo/Lap | | Tumor size (mean or medium, cm) Robo/Lap | Nb LPND | | Follow-up (months) Robo/Lap | Neo-adjuvant therapy (%) Robo/Lap |
|---|---|---|---|---|---|---|---|---|---|---|---|---|---|
| | | | | Mid | Low | | ≤2 | >2 | | Bilateral | Unilateral | | |
| Bae | 60/63 | 21:13/50:24 | - | 20/47 | 14/27 | - | 32/73 | 2/1 | 2.9 (0.5–4.0)/3.1 (1.6–5.0) | 4/6 | 30/68 | - | 88.2/73 |
| Ishizaki | 61/59 | 20:7/22:11 | 21.1/22.8 | - | - | 4/4 | 27/32 | 0/1 | 4.4 (21–80)/4.5 (11–85) | | | - | - |
| Kim | 57/60 | 29:20/21:15 | 23.4/23 | 14/14 | 36/21 | 4.4/5.2 | 46/33 | 4/2 | - | 10/6 | 40/29 | 22.1/28.3 | 82/68.6 |
| Morohashi | 63/63 | 31:6/41:14 | 22.8/22.3 | - | - | 5/5 | - | - | 4 (0–6) / 4 (0–8) | 33/55 | | - | 94.1/78.2 |
| Song | 57.5/60 | 46:24/24:5 | 22.7/22.8 | 19/9 | 51/20 | 4/4 | 66/28 | 4/1 | - | 16/4 | 54/25 | 44.6/39.8 | 87.1/82.8 |
| Zhang | 60.4/58.3 | 54/74 | 22.4/23.8 | - | - | 4.6/4.1 | 82/89 | 13/21 | 4.5 ±1.7/4.4 ± 1.8 | 47/43 | 48/67 | - | 38.9/40 |

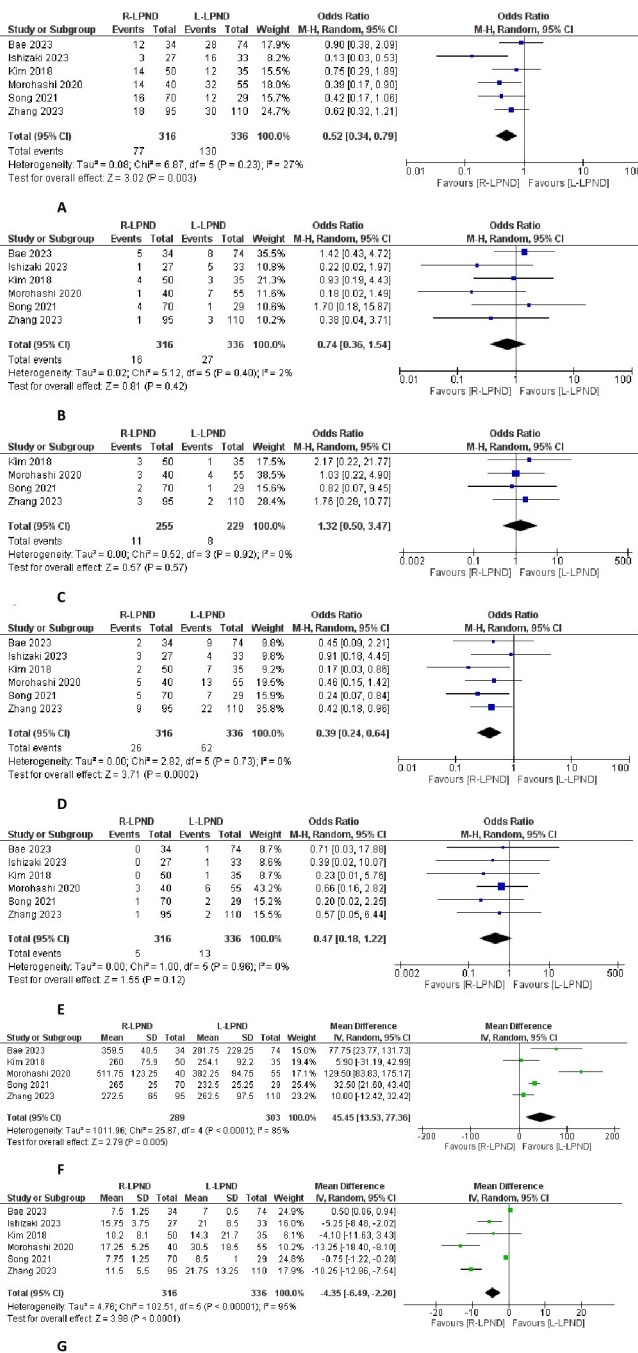

**Fig 2. Forest plots of the postoperative outcomes. A**: Forest plot of the postoperative morbidity. **B**: Forest plot of the anastomotic leak. **C**: Forest plot of the intra-abdominal abscess. **D**: Forest plot of the urinary complications. **E**: Forest plot of the neurological complications. **F**: Forest plot of the operative time. **G**: Forest plot of the hospital stay.

## Operative time

Five studies reported data on operative time [2,16–19]. It was reported in 289 patients in the R-LPND group and 303 patients in the L-LPND group. The operative time was longer in the

R-LPND group (MD = 45.45, 95%CI [13.53 to 77.63], p = 0.005) **(Fig 2F)**. There was high heterogeneity among the studies ($Tau^2$ = 1011.96).

## Hospital stay

Six studies reported hospital stay [2,15–19]. It was reported in 316 patients in the R-LPND group and 336 patients in the L-LPND group. A shorter hospital stay was observed in the R-LPND group (MD = -4.35, 95%CI [-6.49 – -2.20], p<0.001) **(Fig 2G)**. There was low heterogeneity among studies ($Tau^2$ = 4.76).

## LPN dissected

Six studies reported the number of LPNs retrieved [2,15–19]. It was reported in 316 patients in the R-LPND group and 336 patients in the L-LPND group. There were a higher number of harvested lateral pelvic lymph nodes in the R-LPND group (MD = 1.75, 95%CI [0.35 to 3.15], p = 0.01) **(Fig 3A)**. There was low heterogeneity among studies ($Tau^2$ = 1.80).

## LPND time

Three studies reported the LPND time [15,17,19]. It was reported in 162 patients in the R-LPND group and 198 patients in the L-LPND group. There was no significant difference between the two groups (MD = 1.01, 95%CI [-9.00 to 11.02], p = 0.84) **(Fig 3B)**. Moderate heterogeneity was observed among the studies ($Tau^2$ = 65.7).

## Overall recurrence

This outcome has been reported in three studies [2,16,18]. It was reported in 46 of 154 patients in the R-LPND group and 38 of 138 patients in the L-LPND group. There was no significant difference between the two groups (OR = 0.87, 95%CI [0.45 to 1.68], P = 0.69) **(Fig 4A)**. There was low heterogeneity among studies ($Tau^2$ = 0.11).

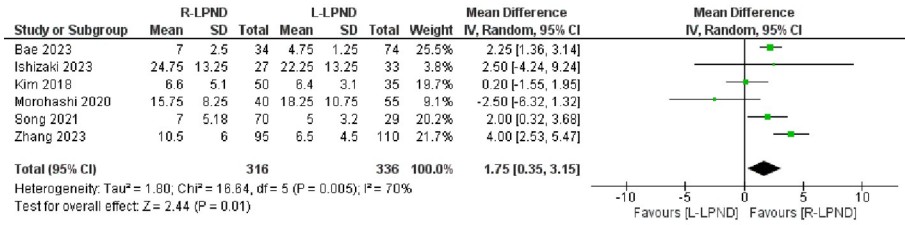

A

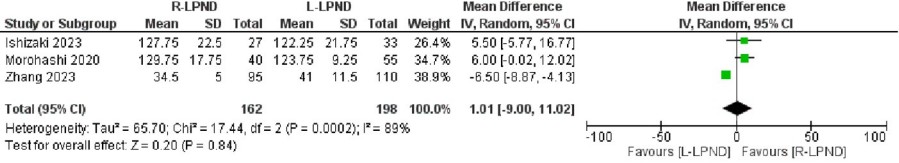

B

**Fig 3. Forest plots of the pathological findings. A:** Forest plot of the lateral pelvic nodes dissected. **B:** Forest plot of the lateral pelvic nodes dissection time.

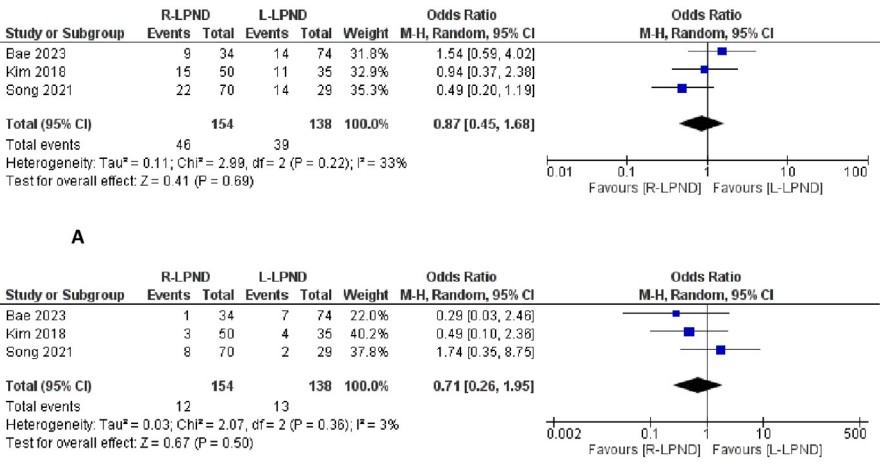

**Fig 4. Forest plots of the oncological outcomes. A**: Forest plot of the overall recurrence. **B**: Forest plot of the local recurrence.

## Local recurrence

This outcome has been reported in three studies [2,16,18]. It was reported in 12 of 154 patients in the R-LPND group and 13 of 138 patients in the L-LPND group. There was no significant difference between the two groups (OR = 0.71, 95%CI [0.26 to 1.95], p = 0.50) **(Fig 4B).** There was low heterogeneity among studies (Tau$^2$ = 0.03).

**Quality assessment of the included studies and reporting of the effects of R-LPND.** MINORS and NOS scores are presented in **Table 1**. A Summary of the evidence is presented in **Table 3**. This review showed that when LPND is performed using the robotic approach, compared to the laparoscopic approach:

- It may reduce morbidity, lower urinary complications, shorter hospital stay, and higher number of harvested lateral pelvic lymph nodes with an increase in operative time.

- We do not know if there were any differences between the two groups in terms of major complications, anastomotic leak, intra-abdominal infection, neurological complications, LPND time, overall recurrence, and local recurrence because the evidence was very uncertain.

## Discussion

This systematic review and meta-analysis concluded that the R-LPND group was associated with lower morbidity, lower urinary complications, shorter hospital stay, and higher harvested lateral pelvic lymph nodes with longer operative time compared to the L-LPND group. There were no differences between the two groups in terms of major complications, anastomotic leak, intra-abdominal infection, neurological complications, LPND time, LPN retrieval, overall recurrence, and local recurrence.

The safety and feasibility of robotic rectal cancer surgery have been well established [22]. LPND in advanced rectal cancer has recently attracted additional attention because of its promising role in local recurrence and improving survival [3,23]. However, for LPND, many

**Table 3. Summary of the retained studies.**

| Outcomes | of participants (studies) Follow-up | Certainty of the evidence (GRADE) | Relative effect (95% CI) | Anticipated absolute effects | |
|---|---|---|---|---|---|
| | | | | Risk with L-LPND | Risk difference with R-LPND |
| Operative time | 592 (5 CCTs) | ⊕⊕◯◯ Low[a] | - | - | MD 45.45 higher (13.53 higher to 77.36 higher) |
| Morbidity | 652 (6 CCTs) | ⊕⊕◯◯ Low[a,b] | OR 0.52 (0.34 to 0.79) | 387 per 1000 | 140 fewer per 1000 (210 fewer to 54 fewer) |
| Major complications | 412 (3 CCTs) | ⊕◯◯◯ Very low[a,b] | OR 1.13 (0.52 to 2.46) | 70 per 1000 | 8 more per 1000 (33 fewer to 87 more) |
| Anastomotic leak | 652 (6 CCTs) | ⊕◯◯◯ Very low[b] | OR 0.74 (0.36 to 1.54) | 80 per 1000 | 20 fewer per 1000 (50 fewer to 38 more) |
| Hospital stay | 652 (6 CCTs) | ⊕◯◯◯ Very low[a,b] | - | - | MD 4.35 lower (6.49 lower to 2.2 lower) |
| Intraabdominal infection | 484 (4 CCTs) | ⊕◯◯◯ Very low[b] | OR 1.32 (0.50 to 3.47) | 35 per 1000 | 11 more per 1000 (17 fewer to 77 more) |
| Urinary complication | 652 (6 CCTs) | ⊕⊕◯◯ Low[b] | OR 0.39 (0.24 to 0.64) | 185 per 1000 | 103 fewer per 1000 (133 fewer to 58 fewer) |
| Neurological complications | 652 (6 CCTs) | ⊕◯◯◯ Very low[b] | OR 0.47 (0.18 to 1.22) | 39 per 1000 | 20 fewer per 1000 (31 fewer to 8 more) |
| LPND time | 360 (3 CCTs) | ⊕◯◯◯ Very low[a] | - | - | MD 1.01 higher (9 lower to 11.02 higher) |
| LPN retrieved | 652 (6 CCTs) | ⊕⊕◯◯ Low[a] | - | - | MD 1.75 higher (0.35 lower to 3.15 higher) |
| Overall recurrence | 292 (3 CCTs) | ⊕◯◯◯ Very low[a,b] | OR 0.87 (0.45 to 1.68) | 283 per 1000 | 27 fewer per 1000 (132 fewer to 116 more) |
| Local recurrence | 292 (3 CCTs) | ⊕◯◯◯ Very low[a,b] | OR 0.71 (0.26 to 1.95) | 94 per 1000 | 25 fewer per 1000 (68 fewer to 74 more) |

*The risk in the intervention group (and its 95% confidence interval) is based on the assumed risk in the comparison group and the relative effect of the intervention (and its 95% CI). CI: Confidence interval; MD: Mean difference; OR: Odds ratio.

GRADE Working Group grades of evidence.

High certainty: We are very confident that the true effect lies close to that of the estimate of the effect.

Moderate certainty: We are moderately confident in the effect estimate: The true effect is likely to be close to the estimate of the effect, but there is a possibility that it is substantially different.

Low certainty: Our confidence in the effect estimate is limited: The true effect may be substantially different from the estimate of the effect.

Very low certainty: We have very little confidence in the effect estimate: The true effect is likely to be substantially different from the estimate of effect.

Explanations.

a. Existing heterogeneity.

b. Small number of event.

colorectal surgeons are still reluctant to perform this procedure using a minimally invasive approach because of its complexity and potential postoperative complications. For TME, robotic surgery has tended to overcome these difficulties with flexible instruments, quality 3D view, stable traction, and endowrists, allowing easier dissection in a deep and narrow pelvis. According to our results, robotics has overcome some of these difficulties with a significant reduction in overall morbidity [24].

Previous studies reported a lower anastomotic leak and intra-abdominal infection in the R-LPND group owing to the ease of incorporating indocyanine green fluorescence imaging [17]. Given the lack of significant differences between the two groups in our systematic review and meta-analysis, we conclude that R-LPND is at least as safe as L-LPND. Because LPND requires only lymphatic tissue dissection, the lateral pelvic vascular and nervous complexes should be preserved, as possible, during skeletonization. In some cases, we found a huge

adhesion around the lymph nodes and we were obliged to scarify certain structures to ensure an oncological safe resection. Furthermore, excessive traction and trembling without direct damage to the major pelvic nerve plexus during dissection can cause urinary complications [25,26]. In our study, we concluded that the incidence of urinary complications was lower in the R-LPND group. Very stable traction and countertraction can cause less injury to the neuronal tissue. Several studies have reported a urinary dysfunction rate of 9.3%–16.7% after LPND [27,28]. This could be due to the resection of the inferior vesical vessels, which seems to increase postoperative urinary dysfunction [29]. Preserving the inferior vesical vessels reduces urinary dysfunction and spares the pelvic splanchnic trunk and plexus [29]. Yamaoka et al. [30], in a retrospective study of 337 patients undergoing TME, found in the multivariate analysis that the laparoscopic approach and open surgery were significantly associated with an increase in postoperative urinary dysfunction. They concluded that robot-assisted surgery was inversely correlated with postoperative early urinary dysfunction, and it may be a better approach to protect urinary function in lower rectal cancer surgery. Song et al. [18], In a retrospective study of 92 patients undergoing TME with selective LPND observed lower urinary retention in the R-LPND group than in the L-LPND group (7.1% vs. 24.1%; p = 0.043).

These findings are highly valuable given the complex anatomy of a region characterized by intricate networks of nerves and blood vessels. Despite comparable specific LPND times, the overall operative time was significantly longer in the R-LPND group than in the L-LPND group. According to this systematic review and meta-analysis, the laparoscopic procedure was 45 min shorter than the robotic procedure. We found high heterogeneity among the different studies when we focused on the mean operative time in the R-LPND group. It ranged between 260 min and 511.75 min. This could be explained by several factors, such as the absence of standardized criteria to measure this outcome and the unavailability of data regarding the learning curves of surgeons [31]. Kim et al. [16] demonstrated the experience of a single highly experienced surgeon. In contrast, Bae et al. [2] reported the experience of five different surgeons. Several studies have compared robotic and laparoscopic TME [32,33]. They reported longer operative times in the robotic surgery group. We believe that this difference could explain the difference between the R-LPND and L-LPND groups in our study because we did not find any difference between the two groups when we compared these two groups in terms of LPND time only. A shorter hospital stay is another additional advantage of the robotic approach. This outcome is essential in reducing infection risk, saving costs, and improving patient experience. However, this difference should be considered with caution in the absence of a clear statement regarding the application of an enhanced recovery protocol.

The retrieval of Laparoscopic Pelvic Nodes (LPN) is a crucial indicator of improved oncological outcomes [34]. Despite the absence of a consensus on the optimal number of harvested LPNs for evaluating pathological outcomes, the quantity retrieved often serves as a measure of the thoroughness of LPND. In our study, a noteworthy observation was the substantially higher number of LPNs harvested in the robotic group than in the laparoscopic group. This disparity in the number of retrieved LPNs can be attributed to two factors. First, the incorporation of fluorescence imaging with infrared optics into the robotic approach provides a distinct advantage. This technology facilitates the identification of metastatic lymph nodes during the robotic procedure, enhancing the surgeon's ability to comprehensively harvest the nodes. Second, robotic en bloc resection of LN263P, LN 263D, and LN 283 using articulated instruments represents a technique that is challenging to replicate with laparoscopic forceps [15,35]. The superiority of the robotic approach in terms of retrieved LPN was not translated to the oncological data assessed in our study, and we found similar overall recurrence and local recurrence. It is worth mentioning that in our study, more patients in the robotic group received neoadjuvant therapy than those in the laparoscopic group. Unlike TME, cumulative

evidence of the oncological safety of robotic LPND is scanty. Only three studies compared the recurrence rates between robotic and laparoscopic LPND. According to the results of a Japanese multicenter study [36] comparing TME with or without LPND, highlighting the reduced local recurrence rate after LPND (7% vs. 13%; p = 0.02), the comparison between robotic and laparoscopic LPND gradually became more critical.

This study had certain limitations that warrant careful consideration. The major limitation of the study is the lack of mature data and paucity of randomized trials, prospective studies, or propensity score matched studies which can control for confounding and selection bias. Additionally, the limited number of studies and participants coupled with missing data contributed to the overall constraints of the study. Notably, the inability to conduct a comparative analysis of the oncological outcomes represents another limitation. In addition, we should mention that all these studies were from Asian countries including patients with a low BMI and this highlighted the interest of additional non-Asian studies for an external validity of these findings. It is crucial to acknowledge that this study, being an inaugural systematic review and meta-analysis examining the two distinct approaches to LPND, may face inherent constraints due to the restricted availability of high-quality prospective studies in this domain.

In conclusion, the robotic approach is safe and feasible for TME with LPND for advanced rectal cancer. It ensures lower morbidity, especially lower urinary complications, with a shorter hospital stay and higher harvested LPN instead of a longer operative time. However, we strongly emphasized that the grade of evidence is low to very low across all measures of outcomes resulting in an inability to draw definite conclusions. Large, multicenter RCTs with longer follow-up would help validate the superiority in terms of morbidity and harvested LPN and will assess a better placement in oncological safety.

## Supporting information

**S1 Checklist. PRSIMA 2020 checklist.**
(DOCX)

## Author Contributions

**Conceptualization:** Mohamed Ali Chaouch, Mohammad Iqbal Hussain, Adriano Carneiro da Costa, Amine Gouader, Eddy Cotte.

**Data curation:** Mohamed Ali Chaouch, Mohammad Iqbal Hussain, Adriano Carneiro da Costa, Bassem Krimi, Amine Gouader, Eddy Cotte, Jim Khan.

**Formal analysis:** Mohamed Ali Chaouch, Mohammad Iqbal Hussain, Adriano Carneiro da Costa, Bassem Krimi, Amine Gouader, Eddy Cotte, Jim Khan, Hani Oweira.

**Funding acquisition:** Mohamed Ali Chaouch, Mohammad Iqbal Hussain, Alessandro Mazzotta, Hani Oweira.

**Investigation:** Mohammad Iqbal Hussain, Adriano Carneiro da Costa, Alessandro Mazzotta, Bassem Krimi, Jim Khan.

**Methodology:** Mohamed Ali Chaouch, Mohammad Iqbal Hussain, Adriano Carneiro da Costa, Alessandro Mazzotta, Amine Gouader, Jim Khan.

**Project administration:** Mohamed Ali Chaouch.

**Resources:** Amine Gouader.

**Software:** Mohamed Ali Chaouch, Eddy Cotte, Hani Oweira.

**Supervision:** Alessandro Mazzotta.

**Validation:** Alessandro Mazzotta, Bassem Krimi, Jim Khan, Hani Oweira.

**Visualization:** Hani Oweira.

**Writing – original draft:** Mohamed Ali Chaouch, Adriano Carneiro da Costa, Bassem Krimi.

**Writing – review & editing:** Eddy Cotte, Jim Khan, Hani Oweira.

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
