## [Decision Letter · Decision Letter 0]

24 Mar 2024

PONE-D-24-01676Robotic Versus Laparoscopic Total Mesorectal Excision with Lateral Lymph Node Dissection for Advanced Rectal Cancer: A Systematic Review and Meta-AnalysisPLOS ONE

Dear Dr. Chaouch,

Thank you for submitting your manuscript to PLOS ONE. After careful consideration, we feel that it has merit but does not fully meet PLOS ONE’s publication criteria as it currently stands. Therefore, we invite you to submit a revised version of the manuscript that addresses the points raised during the review process.

We look forward to receiving your revised manuscript.

Kind regards,

Tsutomu Kumamoto

Academic Editor

PLOS ONE

Journal Requirements:

Additional Editor Comments:

The authors are conducting a very important review in rectal cancer surgery. However, there are several issues.

1. Regarding postoperative complications, more detailed descriptions are required for definitions and grades.

2. Will this review focus on both short-term and long-term results? It would be better to make a clear description of these.

3. There are no RCT or PS matching studies, and the review contains considerable selection bias. Considering this, the review must not mislead the readers.

Reviewers' comments:

Reviewer's Responses to Questions

**Comments to the Author**

1. Is the manuscript technically sound, and do the data support the conclusions?

Reviewer #1: Yes

Reviewer #2: Yes

Reviewer #3: Partly

2. Has the statistical analysis been performed appropriately and rigorously? 

Reviewer #1: Yes

Reviewer #2: Yes

Reviewer #3: Yes

3. Have the authors made all data underlying the findings in their manuscript fully available?

Reviewer #1: Yes

Reviewer #2: Yes

Reviewer #3: Yes

4. Is the manuscript presented in an intelligible fashion and written in standard English?

Reviewer #1: Yes

Reviewer #2: No

Reviewer #3: Yes

5. Review Comments to the Author

Reviewer #1: The authors compared postoperative short course outcomes between robotic and laparoscopic TME+LPLND. The authors concluded that the robotic approach is a safe and feasible alternative in advanced rectal cancer. This is appealing to some physicians, but I have the following issues:

1. As described in discussion, the studies including in this analysis are all retrospective studies, which lead to selection bias.

2. The BMI of the cases is low (mean 21-23), so it should be described that this study is based on a biased case.

3. The definition of each complication is not stated. The grade of the complication should also be described.

4. There are many typographical errors.

Reviewer #2: This study is a meta-analysis conducted to investigate the advantages of robotic surgery over laparoscopic surgery in performing Lateral Pelvic Node Dissection (LPND). Appropriate studies were identified and analyzed. However, I would like to inquire about a few points:

1. The introduction states that while the benefits of postoperative outcomes are identified, the oncological safety has not been clarified. However, the abstract states that this study will clarify postoperative outcomes. If the objective is to compare short-term outcomes, the introduction in the text needs to be refined.

2. In the introduction, the citation for the following sentence needs to be corrected: "The overall risk of recurrence following this strategy increases from 10 (2) to 30% in cases of lateral node involvement (2)."

3. In the introduction, the citation for the following sentence needs to be replaced with a more appropriate one: "In the East, promising results have been demonstrated when LPN metastasis (LPNM) is treated with TME and Lateral Pelvic Node Dissection (LPND) (1)."

4. In the methods, the paragraph on eligibility criteria's studies needs clarification: "We included all randomized controlled trials (RCTs) and controlled clinical trials (CCTs) that compared robotic Total Mesorectal Excision (RTME) with Laparoscopic Pelvic Lymph Node Dissection (LPND), and laparoscopic Total Mesorectal Excision (LTME) with LPND." Please verify if the term 'laparoscopic pelvic lymph node dissection' is incorrectly written instead of 'lateral pelvic lymph node dissection.'

5. To analyze differences in anastomotic leakage, it would be beneficial to add the stoma rate to the results.

6. In the results, it would be advisable to include lymphocele, which is an LPND-specific complication.

7. Regarding the third paragraph of the discussion: "Because LPND requires only lymphatic tissue dissection, the lateral pelvic vascular and nervous complexes should be preserved during skeletonization." I disagree with this statement. I believe that if LPN has to be removed in proximity to certain structures, some vessels and nerves may need to be sacrificed. I would like to hear your opinion on this.

8. The use of abbreviations needs to be systematically reviewed throughout the text. Especially since LPN is described in the fifth paragraph of the discussion, it should have been detailed earlier. And as previously mentioned, the abbreviation "Laparoscopic Pelvic Lymph Node Dissection (LPND)" is inappropriate. There is a need for a clear review and arrangement of the abbreviations used throughout the text.

Reviewer #3: Thank you for the opportunity to review this study. My comments are as follows:

1. This review covers a relevant and interesting topic.

2. The introduction is rather wordy and covers many different concepts. This should be further summarised in shortened paragraphs to discuss the pertinent issues related to the topic at hand.

3. The statistical analysis appears to be sound and well conducted.

4. The major limitation of the study is the lack of mature data and paucity of randomized trials, prospective studies, or propensity score matched studies which can control for confounding and selection bias.

5. Several RCTs and meta-analyses already exist comparing Lap vs Robotic TME. The authors should decide if the study is to compare Lap TME + LPND vs RTME + LPND as currently is, or to focus on the Lap LPND vs Robotic LPND aspect – this is an important distinction as several important quality control and oncological parameters are missing should TME be included, such as rates of complete TME, distal resection margin, CRM positivity, and conversion rates. I suggest that the authors focus on the LPND aspect as this is relatively novel while there is an abundance of data on approaches to TME.

6. Table 2 is quite challenging to compare and interpret – I suggest there be an additional row demonstrating the pooled frequencies and proportions.

7. It should be strongly emphasized that the grade of evidence is low to very low across all measures of outcomes resulting in an inability to draw definite conclusions.

6. PLOS authors have the option to publish the peer review history of their article (what does this mean?). If published, this will include your full peer review and any attached files.

Reviewer #1: No

Reviewer #2: No

Reviewer #3: No

---

## [Author Response · Author response to Decision Letter 0]

31 Mar 2024

Reviewer queries responses 

Editor

Q1. Regarding postoperative complications, more detailed descriptions are required for definitions and grades.

R1. We have added that mortality and morbidity were considered when they occurred within 90 and 30 days after rectal resection, respectively.

Q2. Will this review focus on both short-term and long-term results? It would be better to make a clear description of these.

R2. We have removed the term “long-term” and we have used “oncological” results only to avoid this doubt. 

Q3. There are no RCT or PS matching studies, and the review contains considerable selection bias. Considering this, the review must not mislead the readers.

R3. Thank you for this valuable remark. We have mentioned this detail in the limitation of the study, reported in the risk of bias assessment of the retained studies, and it was a reason to decrease the level of evidence of the different conclusions after the GRADE assessment in the summary of findings. 

Reviewers

Reviewer #1: 

Q1. As described in discussion, the studies including in this analysis are all retrospective studies, which lead to selection bias.

R1. Thank you for this valuable remark. We have mentioned this detail in the limitation of the study, reported in the risk of bias assessment of the retained studies, and it was a reason to decrease the level of evidence of the different conclusions after the GRADE assessment in the summary of findings. 

Q2. The BMI of the cases is low (mean 21-23), so it should be described that this study is based on a biased case.

R2. Yes absolutely. We have also remarked on this point. However, it is explained by the origin of these studies from Asian countries that were well known with their low BMI. We have added this information in the discussion section and we have highlighted its interest for an external validity of our findings. Thank you for this remark. 

Q3. The definition of each complication is not stated. The grade of the complication should also be described.

R3. We have added a definition of the different complications. Unfortunately, we cannot add additional information regarding the grade because it was not reported in the retained studies.

Q4. There are many typographical errors.

R4. We have corrected the different errors. Thank you

Reviewer #2: 

Q1. The introduction states that while the benefits of postoperative outcomes are identified, the oncological safety has not been clarified. However, the abstract states that this study will clarify postoperative outcomes. If the objective is to compare short-term outcomes, the introduction in the text needs to be refined.

R1. We have removed this sentence from the introduction to alleviate this issue. 

Q2. In the introduction, the citation for the following sentence needs to be corrected: "The overall risk of recurrence following this strategy increases from 10 (2) to 30% in cases of lateral node involvement (2)."

R2. We have corrected the references. 

Q3. In the introduction, the citation for the following sentence needs to be replaced with a more appropriate one: "In the East, promising results have been demonstrated when LPN metastasis (LPNM) is treated with TME and Lateral Pelvic Node Dissection (LPND) (1)."

R3. We have added a reference to this sentence. 

Q4. In the methods, the paragraph on eligibility criteria's studies needs clarification: "We included all randomized controlled trials (RCTs) and controlled clinical trials (CCTs) that compared robotic Total Mesorectal Excision (RTME) with Laparoscopic Pelvic Lymph Node Dissection (LPND), and laparoscopic Total Mesorectal Excision (LTME) with LPND." Please verify if the term 'laparoscopic pelvic lymph node dissection' is incorrectly written instead of 'lateral pelvic lymph node dissection.'

R4. We have corrected it. 

Q5. To analyze differences in anastomotic leakage, it would be beneficial to add the stoma rate to the results.

R5. Unfortunately, the stoma rate was not reported in the included studies.

Q6. In the results, it would be advisable to include lymphocele, which is an LPND-specific complication.

R6. This outcome was not reported in the different included studies. 

Q7. Regarding the third paragraph of the discussion: "Because LPND requires only lymphatic tissue dissection, the lateral pelvic vascular and nervous complexes should be preserved during skeletonization." I disagree with this statement. I believe that if LPN has to be removed in proximity to certain structures, some vessels and nerves may need to be sacrificed. I would like to hear your opinion on this.

R7. Yes absolutely. In some cases, we found a huge adhesion of metastasis LN and we are obliged to scarify certain structures. We have added this information to this sentence. 

Q8. The use of abbreviations needs to be systematically reviewed throughout the text. Especially since LPN is described in the fifth paragraph of the discussion, it should have been detailed earlier. And as previously mentioned, the abbreviation "Laparoscopic Pelvic Lymph Node Dissection (LPND)" is inappropriate. There is a need for a clear review and arrangement of the abbreviations used throughout the text.

Q9. We have reviewed all the abbreviations. 

Reviewer #3: 

Q1. The introduction is rather wordy and covers many different concepts. This should be further summarised in shortened paragraphs to discuss the pertinent issues related to the topic at hand.

R1. We have shortened the introduction. 

Q2. The major limitation of the study is the lack of mature data and paucity of randomized trials, prospective studies, or propensity score matched studies which can control for confounding and selection bias.

R2. We have added this information in the limitations of our study. 

Q3. Several RCTs and meta-analyses already exist comparing Lap vs Robotic TME. The authors should decide if the study is to compare Lap TME + LPND vs RTME + LPND as currently is, or to focus on the Lap LPND vs Robotic LPND aspect – this is an important distinction as several important quality control and oncological parameters are missing should TME be included, such as rates of complete TME, distal resection margin, CRM positivity, and conversion rates. I suggest that the authors focus on the LPND aspect as this is relatively novel while there is an abundance of data on approaches to TME.

R3. Thank you for this remark. At the beginning of our study, we performed literature research for all the studies comparing robotic and laparoscopic LPND. We have found studies of urology, digestive disease, and gynaecology. The most relevant issue was that the study population of the PICO that we used was very heterogeneous and it was impossible to pool the outcomes with a huge heterogeneity among the different studies in the majority of the outcomes. We thought that a focus only LPND is an excellent subject of a systematic review and not a meta-analysis. 

Q4. It should be strongly emphasized that the grade of evidence is low to very low across all measures of outcomes resulting in an inability to draw definite conclusions.

R4. Yes, absolutely we have mentioned this information.

---

## [Decision Letter · Decision Letter 1]

29 Apr 2024

PONE-D-24-01676R1Robotic Versus Laparoscopic Total Mesorectal Excision with Lateral Lymph Node Dissection for Advanced Rectal Cancer: A Systematic Review and Meta-AnalysisPLOS ONE

Dear Dr. Chaouch,

Thank you for submitting your manuscript to PLOS ONE. After careful consideration, we feel that it has merit but does not fully meet PLOS ONE’s publication criteria as it currently stands. Therefore, we invite you to submit a revised version of the manuscript that addresses the points raised during the review process.

We look forward to receiving your revised manuscript.

Kind regards,

Tsutomu Kumamoto

Academic Editor

PLOS ONE

Journal Requirements:

**Additional Editor Comments:**

The authors have addressed most of the reviewers’ comments. However, one reviewer has pointed out a few additional changes that should be made. We recommend reviewing the manuscript again.

Reviewers' comments:

Reviewer's Responses to Questions

**Comments to the Author**

1. If the authors have adequately addressed your comments raised in a previous round of review and you feel that this manuscript is now acceptable for publication, you may indicate that here to bypass the “Comments to the Author” section, enter your conflict of interest statement in the “Confidential to Editor” section, and submit your "Accept" recommendation.

Reviewer #2: All comments have been addressed

Reviewer #3: All comments have been addressed

2. Is the manuscript technically sound, and do the data support the conclusions?

Reviewer #2: Yes

Reviewer #3: Yes

3. Has the statistical analysis been performed appropriately and rigorously? 

Reviewer #2: I Don't Know

Reviewer #3: Yes

4. Have the authors made all data underlying the findings in their manuscript fully available?

Reviewer #2: Yes

Reviewer #3: Yes

5. Is the manuscript presented in an intelligible fashion and written in standard English?

Reviewer #2: Yes

Reviewer #3: Yes

6. Review Comments to the Author

Reviewer #2: You've worked hard on the revisions. I think your excellent research results have been sufficiently supplemented. I would like to make one request. It seems necessary to review all the citations in the discussion. There are minor issues such as some references having incorrect numbers or duplications. I recommend reviewing them once again.

Reviewer #3: =====================================================================================================All comments have been addressed

7. PLOS authors have the option to publish the peer review history of their article (what does this mean?). If published, this will include your full peer review and any attached files.

Reviewer #2: No

Reviewer #3: No

---

## [Author Response · Author response to Decision Letter 1]

1 May 2024

Reviewers’ queries responses 

Reviewer #2: 

You've worked hard on the revisions. I think your excellent research results have been sufficiently supplemented. I would like to make one request. It seems necessary to review all the citations in the discussion. There are minor issues such as some references having incorrect numbers or duplications. I recommend reviewing them once again.

Response :

Thank you for the remarks. We have checked all the references to avoid any issues.

---

## [Editor Report · Decision Letter 2]

6 May 2024

Robotic Versus Laparoscopic Total Mesorectal Excision with Lateral Lymph Node Dissection for Advanced Rectal Cancer: A Systematic Review and Meta-Analysis

PONE-D-24-01676R2

Dear Dr. Mohamed Ali Chaouch

We’re pleased to inform you that your manuscript has been judged scientifically suitable for publication and will be formally accepted for publication once it meets all outstanding technical requirements.

Kind regards,

Tsutomu Kumamoto

Academic Editor

PLOS ONE

Additional Editor Comments (optional):

This time, out of three reviewers, one reviewer gave a reject decision, and two reviewer gave a major revision. However, authors have addressed all the points indicated by the two reviewers. There are limitations, but it is a systematic review that will be useful for many colorectal cancer surgeons, and I decided to accept it.

---

## [Editor Report · Acceptance letter]

16 May 2024

PONE-D-24-01676R2 

PLOS ONE

Dear Dr. Chaouch, 

I'm pleased to inform you that your manuscript has been deemed suitable for publication in PLOS ONE. Congratulations! Your manuscript is now being handed over to our production team.

Kind regards, 

on behalf of

M.D., Ph.D. Tsutomu Kumamoto 

Academic Editor

PLOS ONE